# Analysis of the percentages of monocyte subsets and ILC2s, their relationships with metabolic variables and response to hypocaloric restriction in obesity

**Nicté Figueroa-Vega\*, Carolina I. Marín-Aragón, Itzel López-Aguilar, Lorena Ibarra-Reynoso, Elva Pérez-Luque, Juan Manuel Malacara**⊙ *

Department of Medical Sciences, University of Guanajuato, León Campus, León, Gto., México

* jmmalacara@hotmail.com (JMM); ng.figueroa@ugto.mx (NFV)

## Abstract

### Purpose

Obesity results from excess energy intake over expenditure and is characterized by chronic low-grade inflammation involving circulating monocytes (Mo) and group 2 innate lymphoid cells (ILC2s) imbalance. We analyzed circulating Mo subsets and ILC2s percentages and β2-adrenergic receptor (β2AR) expression in lean and obese subjects, and the possible effect of hypocaloric restriction on these innate immune cells.

### Methods

In 139 individuals aged 45 to 57 years, classified in 74 lean individuals (>18.9kg/m$^2$ BMI <24.9kg/m$^2$) and 65 with obesity (n = 65), we collected fasting blood samples to detect Mo subsets, ILC2s number, and β2AR expression by flow cytometry. Lipids, insulin, leptin, and acylated-ghrelin concentrations were quantified. Resting energy expenditure (REE) was estimated by indirect calorimetry. These measurements were repeated in obese subjects after 7-weeks of hypocaloric restriction.

### Results

Non-classical monocytes (NCM) and β2AR expression on intermediate Mo (IM) were increased in obese individuals (p<0.001, in both cases), whereas the percent of ILC2s was decreased (p<0.0001). Stepwise regression analysis showed significantly negative associations of ILC2s with caloric intake, β2AR expression on IM with REE, but a positive relationship between NCM and HOMA-IR. Caloric restriction allowed a significant diminution of NCM and the β2AR expression on IM, as well as, an increase in the percent of classical Mo (CM), and ILC2s. ΔREE was related to ΔCD16$^+$/CD16$^-$ ratio.

**Data Availability Statement:** All relevant data are available within the paper and its Supporting Information files.

**Funding:** This work was supported by the grant CB2014-242065M from Consejo Nacional de Ciencia y Tecnología (CONACYT, México, to Juan M. Malacara). The funders had no role in study design, data collection and analysis, decision to publish, or preparation of the manuscript.

**Competing interests:** The authors have declared that no competing interests exist.

## Conclusions

These findings show that in obesity occur changes in NCM, ILC2s and β2AR expression, which contribute to the low-grade inflammation linked to obesity and might revert with caloric restriction.

## Introduction

In obesity, the imbalance of energy intake/expenditure favors the accumulation of fat [1], usually with chronic low-grade inflammation [2], which has effects on energy metabolism at central and peripheral levels. Inflammation induces fat mobilization and oxidation [2]. Obese individuals are supposed to have lower energy expenditure (REE), however, longitudinal studies show REE increase with obesity [1].

Several immune cell types are key regulators of metabolic homeostasis [3–7]. In white adipose tissue (WAT) from obese subjects change immune cell composition and function with an effect on energy expenditure. Immune cells in the adipose tissue (AT) of lean subjects include T regulatory (Treg) cells, iNKT cells, group 2 innate lymphoid cells (ILC2s), and M2 macrophages; these cell types have distinct roles in the maintenance of AT homeostasis [2]. In obesity, excessive visceral fat accumulation causes adipose tissue dysfunction that leads to chronic-low grade inflammation with adipocyte hypertrophy and hyperplasia, shift from a type 2 to type 1 cytokine-associated inflammatory environment, altered secretion of adipokines (leptin, adiponectin, and other), and changes in proportions and kind of immune cells toward pro-inflammatory monocytes and Th17 lymphocytes, which strongly contributes to obesity-related comorbidities [3–5].

Adipokines are peptide mediators produced by fat cells in AT that exert a powerful influence over immune system [8]. Leptin and adiponectin have effect over functions of dendritic cells, monocytes, neutrophils, and innate lymphoid cells. Leptin, induces satiety, and with other metabolic functions such as regulation of energy expenditure. It also increases phagocytic activity, pro-inflammatory cytokines secretion, and polarization of immune cells toward pro-inflammatory phenotypes [9]. Appetite regulation has a counterregulatory peripheral function in ghrelin, an acylated peptide product of the stomach that induces appetite, and it is also an anti-inflammatory cytokine that suppresses inflammation in obesity [10].

Circulating monocytes (Mo) include three distinct subtypes according to their surface expression of lipopolysaccharide receptor CD14 and FcγIII receptor CD16 [11,12], as follows: classical monocytes (CM; CD14$^{++}$CD16$^-$) account for 80–90% of total monocytes with an anti-inflammatory phenotype. The minor CD16$^+$ Mo subpopulation comprises the remaining 10–15% and is subdivided further into intermediate monocytes (IM; CD14$^+$CD16$^+$) and non-classical monocytes (NCM; CD14$^-$CD16$^{++}$), both with a pro-inflammatory phenotype and are elevated in chronic inflammatory and metabolic diseases [12–18]. Changes in monocytes subsets CD14$^{++}$CD16$^-$, CD14$^+$CD16$^+$ and CD14$^-$CD16$^{++}$ have been described after dietary interventions in obese individuals [19–22]. Monocytes can migrate toward AT and endothelium, where they become converted into macrophages. Therefore, it is important to detect these subpopulations in peripheral blood for the evaluation the chronic low-grade inflammation and regulation of energy in obesity.

Innate lymphoid cells (ILCs) comprise three subpopulations: ILC1, ILC2 and ILC3. Recent evidences indicate that ILCs are involved in the progression of several metabolic diseases. These cells promote obesity, and are involved in adipose tissue inflammation [23,24].

Nevertheless, group 2 innate lymphoid cells (ILC2s), anti-obese immune regulators in AT, secrete anti-inflammatory cytokines, promote polarization into M2 macrophages, eosinophil regulating adaptive immunity, limiting obesity and promoting the browning of WAT [25]. ILC2s synthesize IL-5 and IL-13, cytokines implicated in browning of WAT (thermogenesis) [26]. However, the mechanisms by which ILC2s regulate AT homeostasis are incompletely defined. ILCs has been characterized in AT but information about their presence in circulation is not available. Therefore, it is needed a new approach for their study and establish their connections with energy metabolism. Due to the complexity for the isolation of ILC2 cells from human AT, peripheral bloods specimens should be used to monitor ILC2s.

Sympathetic and parasympathetic systems converge in the activation of β2-adrenoceptors on immune cells to control systemic inflammation allowing the crosstalk between nervous, endocrine and immune systems [27]. β2AR are found in inflammatory cells such as mast cells, monocytes, eosinophils, T-lymphocytes, and neutrophils. β2AR expression on Mo and their activation has usually anti-inflammatory effect [28].

The aims of this work were to determine circulating Mo subsets and ILC2s and β2AR expression in obese and lean subjects, and to evaluate the effect of moderate diet restriction on these cells, and the interaction of resting energy expenditure (REE), and metabolic variables in subjects with obesity.

## Materials and methods

### Participants

We recruited 139 participants of 20 to 50 years old from León, Mexico, classified in two groups, 65 with obesity (BMI $\geq$30 (kg/m$^2$), and 74 lean subjects (BMI >18.9kg/m$^2$ to $\leq$24.9kg/m$^2$).

Participants did not have clinical evidence of chronic or infectious diseases and were not taking anxiolytics, antidepressants, β-blockers, Ca$^{++}$ channel blockers, antibiotics, or hypnotic drugs. Women did not receive hormone therapy in the previous six months, and were not pregnant, or lactating. No volunteer with smoking habit or habitual alcohol consumption was included.

### Data collection

We collected age, weight measured with a roman type scale, and height with a Stadiometer (SECA 216). BMI and percentage weight loss (%WL) were calculated. Blood pressure was measured in sitting position after 5 min rest. Physical activity was evaluated using the International Physical Activity Questionnaire (IPAQ) [the short, last seven days self-administered version of IPAQ from the 2000/01 Reliability and Validity Study]. Collection of all measurements was performed in basal state and at the end of the diet period in individuals with obesity (Fig 1).

### Resting energy expenditure (REE)

REE was estimated by indirect calorimetry (IC) with a Fitmate device (Wellness Technology, Cosmed, USA), calibrated before each assessment, following the manufacturer's specifications. For evaluation, subjects abstained from physical exercise, drinks with coffee or black tea the previous 24 h. The assessment was carried out after 8–10 h fasting in a controlled environment with room temperature 21–24˚C, with low light and no noise. During the 15 minutes of measurement, subjects were awake and in supine position, measurements on the first 10 min were discarded to improve stability.

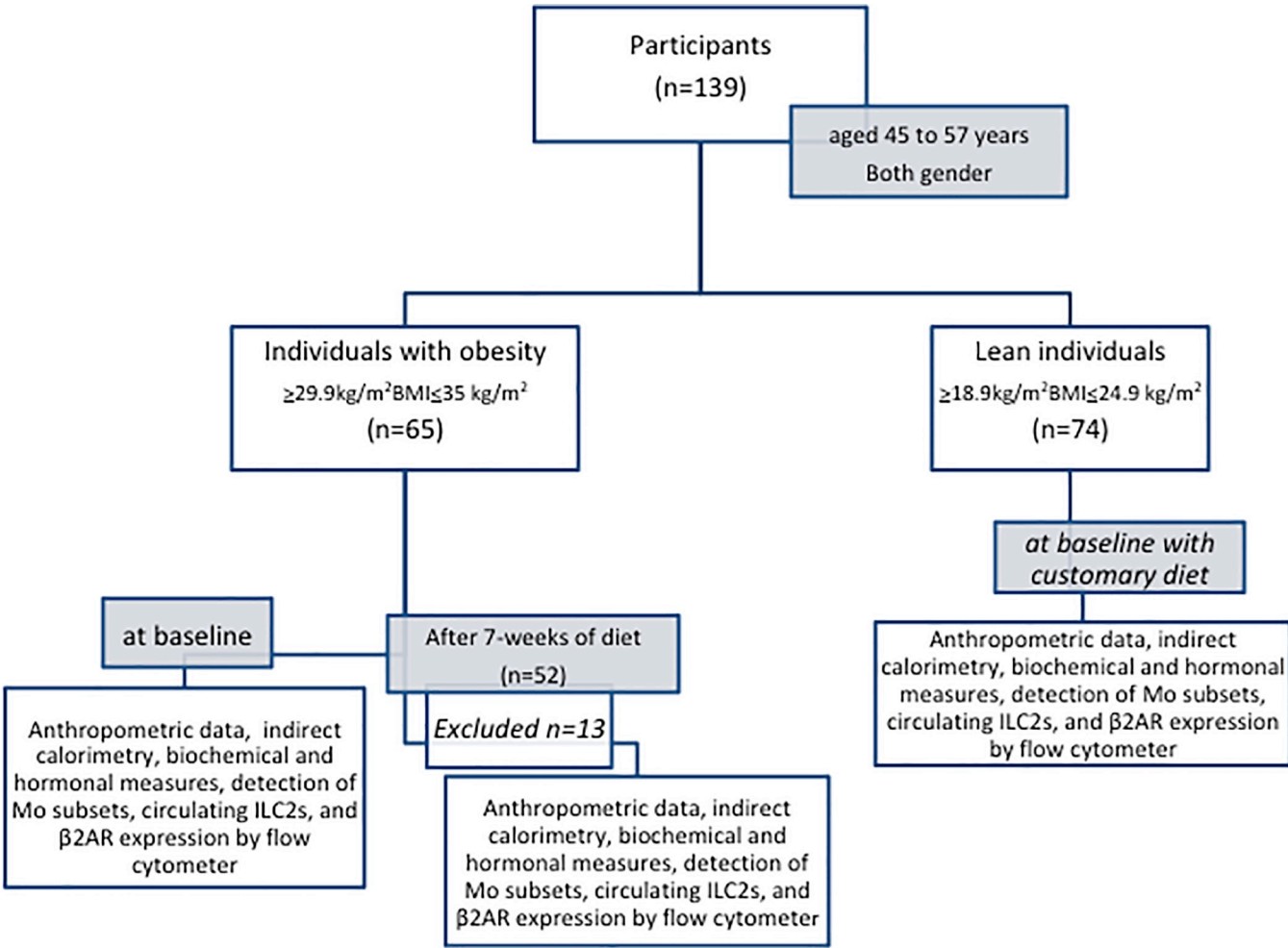

**Fig 1. Flow chart for the clinical procedure.** Diagram illustrates the sequence of procedures carried out on the study for selection and classification of participants in both groups, assignation of diet in individuals with obesity, and measures.

## Samples

Peripheral blood samples were obtained after 8–10 h overnight fasting. Serum was separated and stored at -80˚C until use. Peripheral mononuclear cells were isolated from heparinized blood. The samples were obtained before and after intervention for measurement of all variables.

## Dietary intervention

Personalized caloric restriction was prescribed for 7-weeks, reducing 580 kcal of current intake of each individual with obesity. Dietary intake was designed in kilocalories/day with the following percentage of macronutrients: 55%–60% carbohydrates, 15% proteins and 30% lipids, as recommended for obese individuals, by the North American Association for the Study of Obesity (NAASO) and the National Heart, Lung, and Blood Institute (NHLBI) 2000.

Dietary intake was evaluated with 24-hour recollection (two weekdays and one weekend), at baseline and on each of the three weeks of the intervention. We analyzed 24-hours recollections with the Food Processor SQL-Nutrition Software, to quantify energy intake (Kcal) and

type of macronutrients. Adherence to diet was considered satisfactory when energy consumption was ±80% of the amount prescribed. Normal weight subjects continued with their customary diet.

## Metabolic and hormonal measurements

Serum glucose and lipid profile were measured using enzymatic methods with a chemical analyzer (Microlab 300, ELITECH group, Vital Scientific, Netherlands). LDL-cholesterol was calculated by the Friedewald´s formula [29]. Non-HDL cholesterol was also calculated. Circulating acylated-ghrelin (MyBiosource, CA, USA), insulin (ALPCO, Salem, NH), and leptin (ALPCO) concentrations were measured by ELISA. All analyses were carried out by duplicate.

## Cell isolation

Peripheral blood mononuclear cells (PBMC) were isolated by Ficoll-Hypaque (1.077 g/ml; Sigma-Aldrich) gradient. Viability was examined using trypan blue exclusion.

## Identification of monocytes subsets

PBMC were stained with anti-CD14 mAb conjugated to Fluorescein isothiocyanate (FITC) and–CD16 mAb conjugated to Allophycocyanin (APC) (both purchased from BD Biosciences, San Jose, CA, USA). To set the gates, control isotype antibodies were employed, an anti-mouse Ig kappa chain-FITC (BD Pharmingen), and an anti-mouse IgG1-APC (BioLegend). Moreover, monocytes were gated on the basis of their size (forward scatter) and complexity (side scatter), and the monocyte subsets were identified by the levels of expression of CD14 and CD16. Fifty thousand events were acquired for each sample on a FACSCanto II (two-laser, six-color configuration) with the FACSDiva 6.1.3 software (BD Biosciences). The results are shown as percentage of each subpopulation. Nomenclature of monocyte subsets followed the recommendations of the Nomenclature Committee of the International Union of Immunological Societies. We carried out daily routine quality control tests with Cytometer Setup & Tracking Beads (BD Biosciences) in accordance with the manufacturer's instructions.

## Evaluation of β2AR expression

The level of expression of β2AR was analyzed by a three-color flow cytometry assay. Briefly, cells were incubated with an anti-β2AR (IgG1, clone 6H8; Abcam) mAb, followed by a goat anti-mouse IgG polyclonal Ab labeled with PE (Abcam). Then, cells were stained for CD14 and CD16, as stated above. Finally, cells (at least 50,000, and gated according to their FS and SS characteristics) were analyzed in a FACSCanto II (two-laser, six-color configuration) flow cytometer with the FACSDiva 6.1.3 software (BD Biosciences). Results were shown as the median intensity of fluorescence (MFI) of positive cells.

## Phenotypic characterization of ILC2s

PBMC were stained with the following antibody mix: FITC-conjugated anti-human CD2, CD3, CD14, CD16, CD19, CD56, and CD235a (negative lineage cocktail), Phycoerytrin-(PE)-conjugated anti-CRTH2 (chemoattractat receptor-homologous molecule expressed on Th2 cells), Peridinin-Chlorophyll Protein Complex-(PerCP-)-Cyanin-5-conjugated anti-human CD127 (IL-7R) and Allophycocyanin-(APC-)-conjugated anti-human IL-5 (all purchased from BD) in presence or not of PMA/Io (Sigma-Aldrich, USA) and a Golgi inhibitor (Brefeldin A; BD Biosciences). Then, lymphocytes were gated according to their FSC/SSC

characteristics, and lineage-negative (Lin⁻) were selected and analyzed for the expression of CD127 and CRTH2. Finally, These Lin-CD127-CRTH2+ were analyzed for the intracellular expression of IL-5. According to this analysis, Lin⁻CD127⁻CRTH2⁺IL-5⁺ events were considered as ILC2 cells and cells were gated by running fluorescence minus one (FMO) control tubes. Cells were analyzed in a FACSCanto II (two-laser, six-color configuration) flow cytometer with FACSDiva 6.1.3 software (BD Biosciences).

## Statistical analysis

Descriptive statistics was used to show the characteristics of subjects, normality was assessed with the Kolmogorov-Smirnov test. Data are shown as the means ± SD. We compared groups of lean *vs* obese subjects using the Student's T test for independent variables. Changes in variables before and after diet were examined with paired Student´s T test.

We examined factors associated with monocyte subsets percent, ILC2s percent, and β2AR expression using multiple forward stepwise regression analysis testing as candidate regressors: REE, BMI, mean arterial tension, HDL-C, non-HDL-C, triglycerides, acylated-ghrelin, leptin, HOMA-IR, and caloric intake values; and as confounding factors: gender and age.

For analyses we used the Statistica 5.0 (Stat Soft Inc., Tulsa, OK), and Prism 7.0v (Graph-Pad) softwares. $p < 0.05$ was considered statistically significant.

## Ethical approval

All individuals gave written informed consent to participate in the study. The study was carried out according to the ethical standards of the Declaration of Helsinki in 1983 and in agreement with the Good Clinical Practice guidelines. The Institutional Ethics Committee of the University of Guanajuato approved the study with number CIBIUG No. 017/2015.

## Results

We studied 65 unrelated obese subjects (34 women and 31 men) of 35.4±8.0 years old, and 74 lean subjects (51 women and 23 men) aged 30.1±7.0 years. No gender differences were observed between groups of subjects with and without obesity.

The comparison of characteristics among lean and obese subjects is shown in Table 1. REE was significantly higher in obese than lean subjects, but glucose and acylated ghrelin concentrations were not different between both groups. As expected, individuals with obesity had higher blood pressure and insulin resistance.

We identified and quantified the three populations of circulating monocytes to estimate systemic inflammation (Fig 2). Both non-classical monocytes (NCM) percentage and CD16⁺/CD16⁻ ratio were increased in obese subjects suggesting a deregulation between pro-inflammatory and anti-inflammatory subsets (Table 1).

## Increased β2AR expression on intermediate monocytes

The β2AR expression on NCM (CD14⁻CD16⁺⁺) and CM (CD14⁺⁺CD16⁻) was not different within the groups of obese and non-obese subjects. Yet, in obese subjects, the IM (CD14⁺CD16⁺) expressed significantly more β2AR ($p < 0.001$) than in non-obese subjects (Table 1 and Fig 3).

**Table 1. Comparison of basal characteristics among individuals.**

|  | Lean subjects Mean±S.D. (n = 74) | Obese subjects Mean±S.D. (n = 65) | t | *p-value* |
|---|---|---|---|---|
| Age (yr) | 30.1 ± 7 | 35.4 ± 8 | -4.0 | <0.001 |
| Weight (kg) | 63.5 ± 9.3 | 89.5 ± 15.3 | 13.4 | <0.0001 |
| BMI (kg/m$^2$) | 22.8 ± 1.9 | 34.1 ± 4.7 | 19.1 | <0.0001 |
| Mean arterial tension (mmHg) | 84 ± 7 | 90 ± 6.5 | -5.6 | <0.0000001 |
| Resting energy expenditure (kcal/day) | 1530 ± 286 | 1818 ± 380 | -5.09 | <0.0000011 |
| Glucose (mg/dl) | 89 ± 12 | 92 ± 13 | -1.4 | 0.17 |
| HDL-cholesterol (mg/dl) | 62±11 | 52±12 | 5.1 | <0.000001 |
| Non-HDL-cholesterol (mg/dl) | 112±32 | 140±40 | -4.5 | <0.000016 |
| Triglycerides (mg/dl) | 112±60 | 178±196 | -2.8 | <0.0067 |
| Creatinine (mg/dl) | 0.9±0.2 | 0.9±0.2 | -0.6 | 0.52 |
| HOMA-IR | 1.9±0.8 | 3.6±2.2 | -6.2 | <0.00000001 |
| Acylated-ghrelin (µg/ml) | 125.3±100.4 | 104.0±47.6 | 1.4 | 0.16 |
| Leptin (pg/ml) | 21.7±18.2 | 43.4±24.5 | -5.9 | <0.00000002 |
| Insulin (µUI/ml) | 8.6±3.4 | 15.4±8.8 | -6.2 | <0.00000001 |
| *Monocyte subsets* |  |  |  |  |
| Non-classical monocytes (NCM; CD14$^-$CD16$^{++}$) (%) | 4.0±2.5 | 7.2±5.7 | -4.4 | <0.000022 |
| Intermediate monocytes (IM; CD14$^+$CD16$^+$) (%) | 6.50±4.3 | 7.4±5.0 | -1.1 | 0.26 |
| Classical Monocytes (CM; CD14$^{++}$CD16$^-$) (%) | 65.0±11.7 | 62.7±15.1 | 1.0 | 0.31 |
| CD16$^+$/CD16$^-$ ratio | 0.2±0.1 | 0.3±0.2 | -3.1 | <0.0024 |
| *Innate lymphoid cells* |  |  |  |  |
| ILC2s (%) | 11.3±6.7 | 2.9±2.5 | 4.65 | <0.0001 |
| *β2AR expression within monocyte subsets* |  |  |  |  |
| β2AR expression on NCM (MFI) | 10191±4738 | 11912±9718 | -1.0 | 0.33 |
| β2AR expression on IM (MFI) | 25966±15911 | 28370±17169 | -0.7 | <0.001 |
| β2AR expression on CM (MFI) | 13363±5846 | 15877±12429 | -1.1 | 0.50 |

Data are shown as the means ± SD. Differences between groups were evaluated with Student's T test. $p<0.05$ was considered statistically significant.

BMI, body mass index; HOMA-IR, homeostatic model assessment- insulin resistance; β2AR, beta-2 adrenergic receptor; NCM, non-classical monocytes (CD14$^-$CD16$^{++}$); IM, intermediate monocytes (CD14$^+$CD16$^+$); CM, classical monocytes (CD14$^+$CD16$^-$); ILC2s, group 2 innate lymphoid cells; MFI, mean fluorescence intensity.

## Quantification of circulating group 2 innate lymphoid cells (ILC2s)

We defined ILC2s subset as Lin$^-$CD127$^+$CRTH2$^+$IL-5$^+$ cells (Fig 4A). Flow cytometric analysis revealed the presence of ILC2s in peripheral blood, which are significantly diminished in obese subjects in comparison with lean subjects ($p<0.0001$) (Table 1 and Fig 4B).

## Relationship between monocytes subsets, ILC2s percentages and β2AR expression with anthropometric and metabolic characteristics at baseline

We examined the association of ILC2s, monocyte subsets, and β2AR expression on Mo, with anthropometric and metabolic factors in the whole group of study, using a multiple regression model (Table 2).

Non-classical monocytes (CD14$^-$CD16$^{++}$) correlated positively with HOMA-IR, but negatively with acylated-ghrelin levels. Intermediate monocytes (CD14$^+$CD16$^+$) were related to caloric intake. CD16$^+$/CD16$^-$ ratio correlated positively with BMI, and inversely with leptin levels. ILC2s percentage was associated positively with HDL-C levels but inversely with caloric intake.

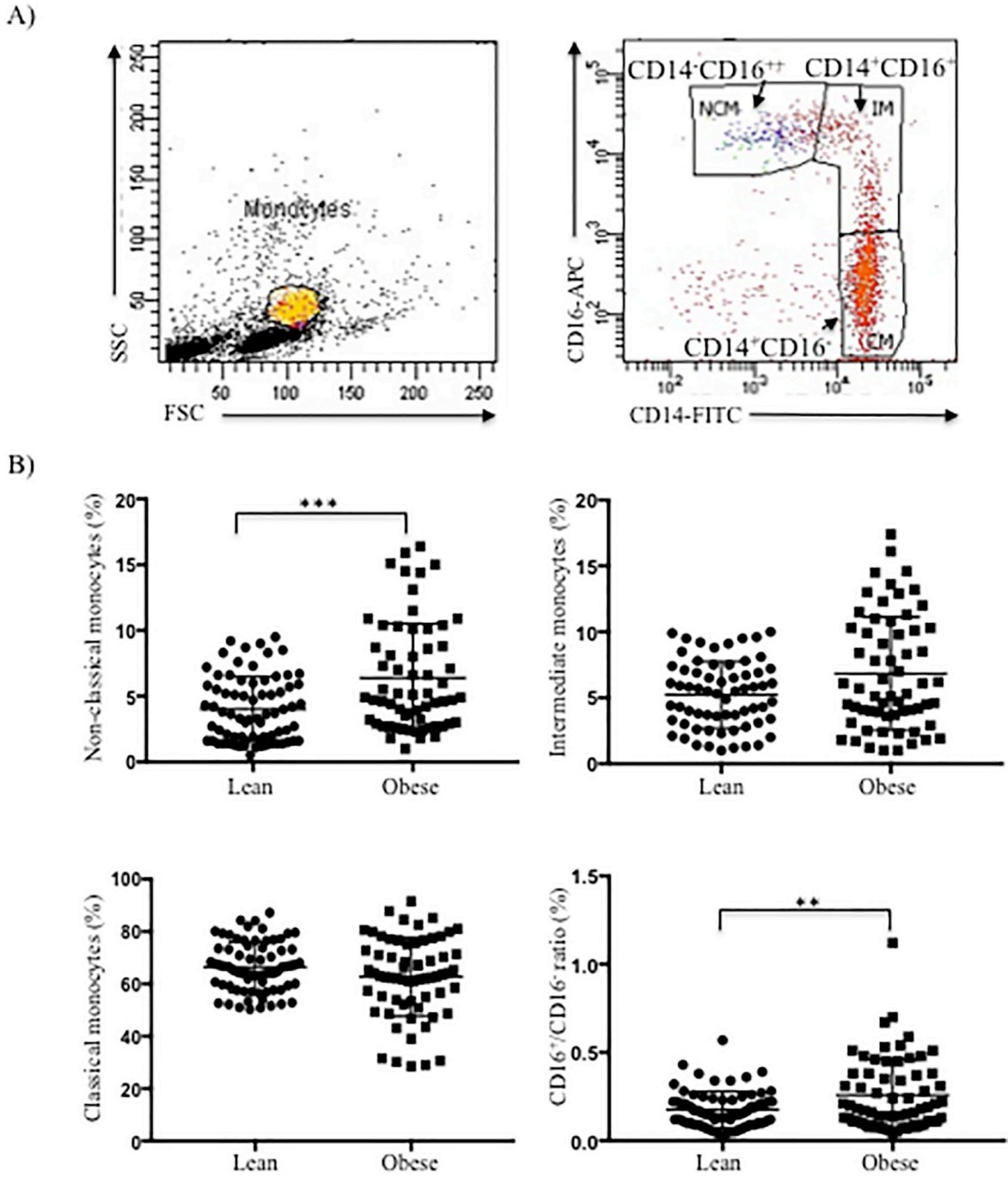

**Fig 2. Number of Mo subsets in lean and obesity state.** PBMC were isolated and then surface-stained with mAbs agaomts CD14 labeled with FITC and CD16 labeled with APC. A) Gating strategy to identify the three monocyte subsets based on relative CD14 and CD16 expression. Flow cytometry dot plot showing the gating of the classical (CM; CD14$^{++}$CD16$^-$), intermediate (IM; CD14$^+$CD16$^+$) and non-classical monocyte (NCM; CD14$^-$CD16$^{++}$) subsets. B) Scatter plots represent the percentage of each Mo subset in both groups. Normal-weight individuals (circles) and subjects with obesity (squares). *$p < 0.05$, **$p < 0.01$, ***$p < 0.001$.

On the other hand, β2AR expression on NCM was associated with HDL-C, leptin levels, and mean arterial tension. There were positive relationships between β2AR expression on IM with HDL-C, and BMI, but negatively with REE. β2AR expression on CM was associated with HDL-C levels, BMI, and age.

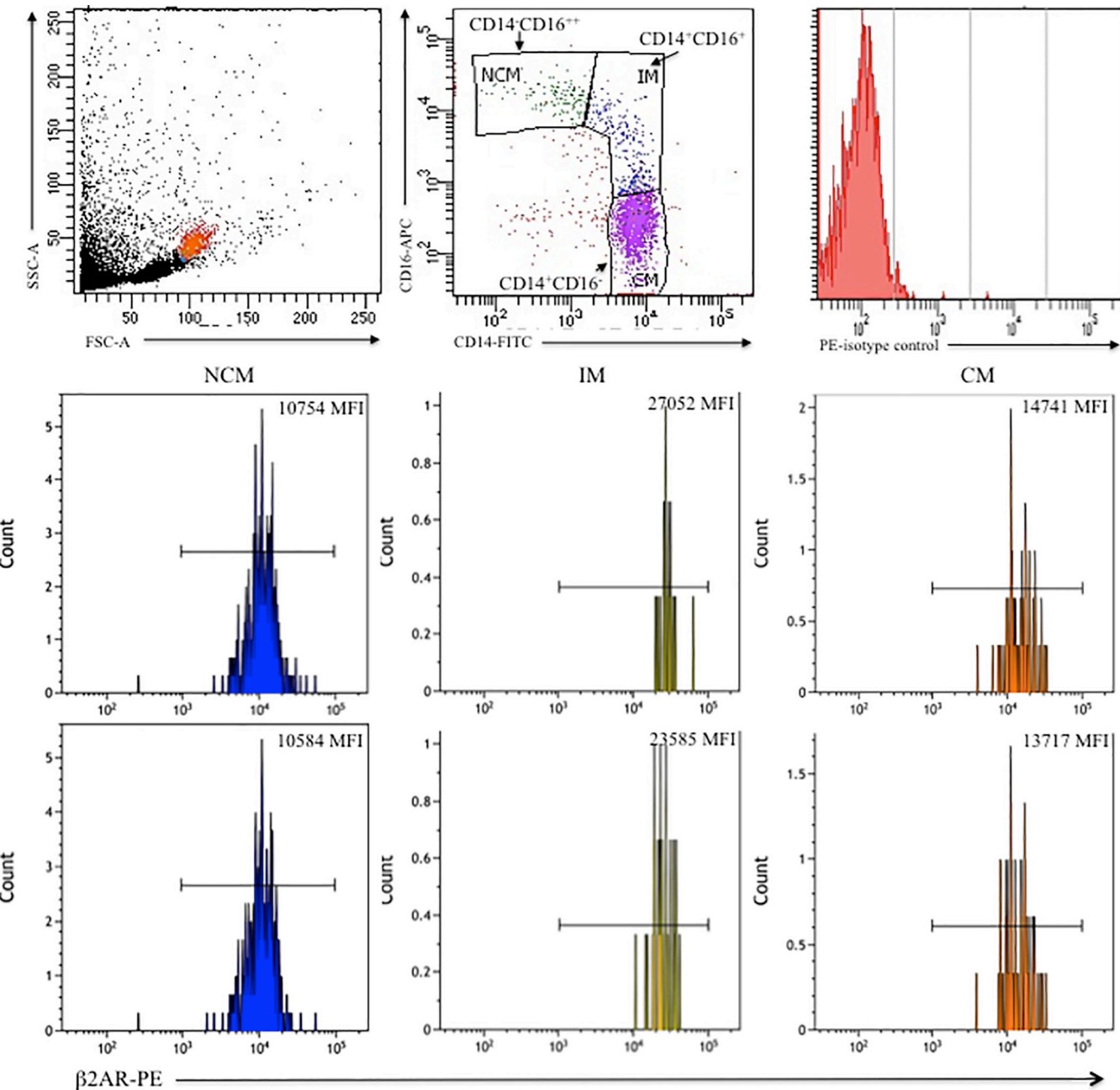

**Fig 3. β2AR expression among Mo subsets.** PBMC were incubated for 1 h with anti-β2AR polyclonal Ab and next stained with a secondary F´(ab)2 PE mAb. Next, PBMC were labeled with anti-CD14-FITC and–CD16-PE mAbs. A) Representative dot plot depicts FSC and SSC to identify monocytes. B) PE-isotype control is shown in a histogram. C) Histograms correspond to β2AR staining within the each Mo subset as follows: the expression of β2AR on CD14$^-$CD16$^{++}$ or NCM (left panels), on CD14$^+$CD16$^+$ or IM (middle panels), and on CD14$^{++}$CD16$^-$ or CM (right panels). MFI is indicated. Results shown are from a representative lean individual (top panels) and a subject with obesity (bottom panels). MFI = mean fluorescence intensity.

After testing for confounding factors, the models did not change.

## Effects of hypocaloric diet over monocyte subsets, circulating ILC2s and β2AR expression

To determine the effect of the hypocaloric diet on the monocyte subsets, ILC2s, and β2AR expression, we compared values before and 7-weeks after diet. Thirteen individuals with

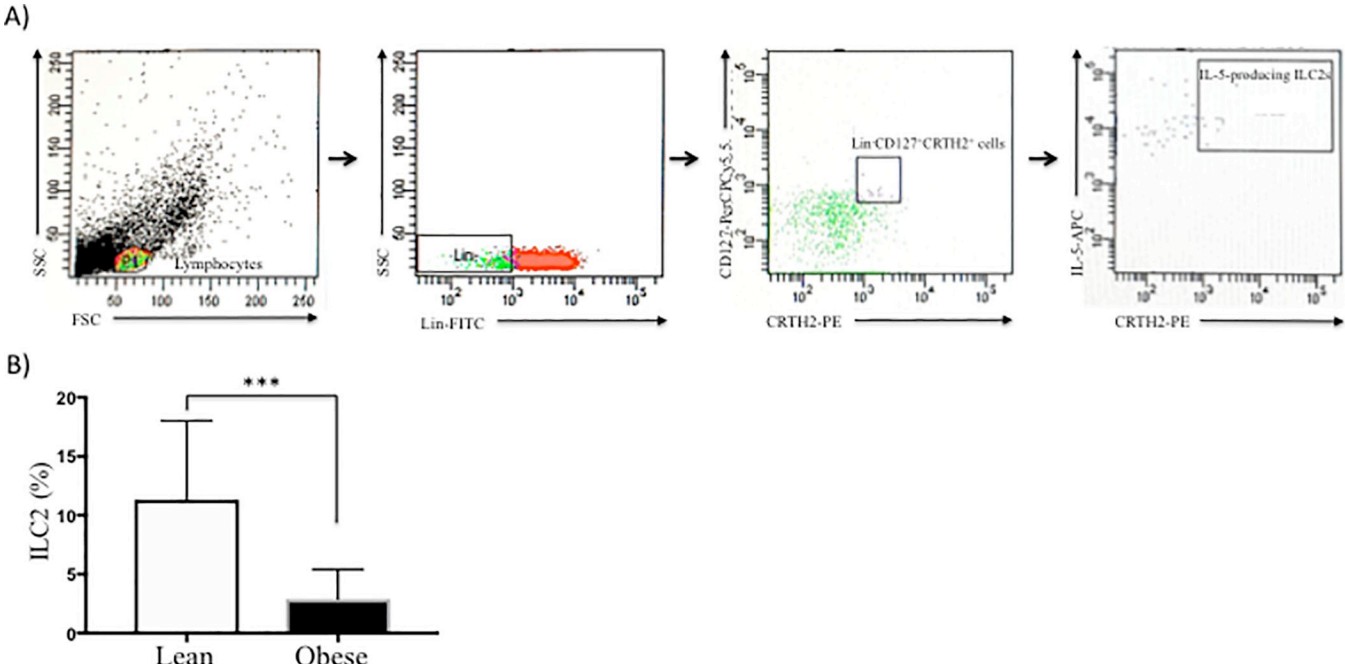

**Fig 4. Isolation and characterization of circulating ILC2s in peripheral blood.** PBMC were stimulated for intracellular cytokine production with PMA/Io and Golgi inhibitors (Brefeldin A) for 5 h at 37˚C/5% $CO_2$, washed and next the cell were surface stained with conjugated antibodies against lineage-FITC, CD127-PerCPCy5.5, and CRTH2-PE. Then fixed, permeabilized and labeled with IL-5 conjugated to APC mAb. Gating strategy to isolate ILC2s population was as follow: we used a lineage cocktail of antibodies to CD2, CD3, CD14, CD16, CD19, CD56, and CD235a to identify T cells, monocytes, neutrophils, B cells, NK cells, mast cells and basophils. Then, we gated Lin⁻ cells, which are negative for these lineage markers, and were further subdivided based on CD127 and CRTH2 expression (third dot plot). Finally, synthesis of IL-5 on Lin⁻CD127⁺CRTH2⁺ population was gated, and we identified this subpopulation such as ILC2. A) Representative FACS plots of strategy for selection of Lin⁻CD127⁺CRTH2⁺IL-5⁺ cells, called ILC2s from a representative individual with obesity. B) Percentages (mean±S.D.) of ILC2 from normal-weight subjects (open bars) and individuals with obesity (solid bars) are shown. \*\*$p<0.01$. PMA, phorbol 12-myristate 13-acetate; Io, ionomycin; CD127, interleukin-7 receptor; CRTH2, chemoattractant receptor-homologous molecule expressed on Th2 cells; mAb, monoclonal antibody.

obesity were excluded for unsatisfactory adherence to diet, resulting in a sample size of 52 subjects. At the end of intervention, subjects showed a mean weight loss of 4.0 kg, corresponding to a 4.5%. REE did not change, but triglycerides and insulin resistance decreased significantly (Table 3).

We also evaluated possible modification of the abundance of Mo subsets and ILC2s and β2AR expression on monocytes after diet (Table 3 and Fig 5). We found that NCM (CD14⁻CD16⁺⁺) decreased and CM (CD14⁺CD16⁻) increased significantly (Fig 5A and 5B). Yet, IM had a non-significant decrease (Fig 5C). There was also a significant diminution in the CD16⁺/CD16⁻ ratio (Fig 5D). Furthermore, circulating ILC2s increased (Fig 5F), and β2AR expression on IM significantly decreased (Fig 5E).

## Associations of changes in monocyte subsets, ILC2s percentage and β2AR expression after dietary intervention

Table 4 shows the changes in NCM (CD14⁻CD16⁺⁺), IM (CD14⁺CD16⁺), and CD16⁺/CD16⁻ ratio after diet, negatively associated with ΔHDL-C and Δleptin levels. The change in CM (CD14⁺CD16⁻) was positively associated with ΔHDL-C concentrations. Interestingly CD16⁺/CD16⁻ ratio showed associations positive with ΔREE but negative with ΔBMI. The ΔCM (CD14⁺CD16⁻) was positively associated only with ΔHDL-C. The changes in ILC2s were negatively correlated with ΔHOMA-IR. The changes in the β2AR expression on NCM and CM

**Table 2. Relationships of monocytes subsets with anthropometric, and metabolic features in basal state.**

| Dependent variable | Regressors | β±S.E. | T | p-level |
|---|---|---|---|---|
| *Non-classical monocytes* (NCM; CD14⁻CD16⁺⁺) Adjusted $R^2$ = 0.135 | | | | |
| | Intercept | | 4.97 | <0.000002 |
| | HOMA-IR | 0.33±0.08 | 4.26 | <0.000037 |
| | Acylated-ghrelin | -0.16±0.08 | 1.98 | <0.04 |
| *Intermediate monocytes* (IM; CD14⁺CD16⁺) Adjusted $R^2$ = 0.127 | | | | |
| | Intercept | | 2.62 | <0.0098 |
| | Caloric intake | 0.17±0.08 | 2.00 | <0.047 |
| *CD16⁺/CD16⁻ ratio* Adjusted $R^2$ = 0.080 | | | | |
| | Intercept | | 0.27 | 0.78 |
| | BMI | 0.35±0.09 | 3.73 | <0.00028 |
| | Leptin | -0.19±0.09 | -2.06 | <0.041 |
| *ILC2s* Adjusted $R^2$ = 0.098 | | | | |
| | Intercept | | 4.06 | <0.000082 |
| | Caloric intake | -0.24±0.08 | -2.87 | <0.0046 |
| | HDL-C | 0.16±0.08 | 1.97 | <0.05 |
| *β2AR expression on NCM* Adjusted $R^2$ = 0.119 | | | | |
| | Intercept | | -2.37 | <0.019 |
| | Mean arterial tension | 0.28±0.08 | 3.35 | <0.001 |
| | Leptin | 0.22±0.08 | 2.54 | <0.012 |
| | HDL-C | 0.21±0.08 | 2.50 | <0.013 |
| | HOMA-IR | -0.18±0.09 | -2.01 | <0.045 |
| *β2AR expression on IM* Adjusted $R^2$ = 0.322 | | | | |
| | Intercept | | -0.70 | 0.48 |
| | HDL-C | 0.42±0.08 | 5.13 | <0.000001 |
| | BMI | 0.41±0.09 | 4.52 | <0.00001 |
| | REE | -0.25±0.08 | -2.92 | <0.004 |
| *β2AR expression on CM* Adjusted $R^2$ = 0.360 | | | | |
| | Intercept | | -1.38 | <0.016 |
| | HDL-C | 0.30±0.08 | 3.45 | <0.0007 |
| | BMI | 0.24±0.09 | 2.63 | <0.009 |
| | Age | 0.21±0.08 | 2.45 | <0.015 |
| | Non-HDL-C | -0.17±0.08 | -2.08 | <0.039 |

The associations were evaluated by multiple regression. $p < 0.05$ was considered statistically significant.

BMI, body mass index; EE, energy expenditure; β2AR, beta-2 adrenergic receptor; NCM, non-classical monocytes (CD14⁻CD16⁺⁺); IM, intermediate monocytes (CD14⁺CD16⁺); CM, classical monocytes (CD14⁺CD16⁺); ILC2s, group 2 innate lymphoid cells.

were associated negatively with ΔBMI and positively with Δleptin in both cases. Also the Δβ2AR expression on IM was associated negatively with ΔBMI. The Δβ2AR expression on NCM was positively related with ΔHDL-C, but negatively with ΔHOMA-IR.

## Discussion

In this work we studied the three types of circulating monocytes and ILC2s, and the elevation of β2AR expression on intermediate monocytes as estimators of chronic low-grade inflammation in individuals with obesity, We analysed its associations with energy expenditure, anthropometry, hormonal and metabolic variables before and after of dietary restriction. The importance of chronic low-grade inflammation in obesity is well described, implicating

**Table 3. Changes in anthropometric, biochemical, and hormonal measures, monocyte subsets, ILC2 and β2AR expression after caloric restriction.**

| Variable | Before CR Mean±S.D. | After CR Mean±S.D. | Δ | t | p-value |
|---|---|---|---|---|---|
| Weight (kg) | 90.1±14.5 | 86.0±14.8 | -4.05 | 16.7 | <0.0001 |
| BMI (kg/m$^2$) | 33.9±4.7 | 32.3±4.7 | -1.5 | 17.8 | <0.0000000001 |
| Mean arterial tension (mmHg) | 90.1±6.4 | 86.2±5.4 | -3.8 | 3.9 | <0.00022 |
| REE (kcal/day) | 1768±377 | 1722±329 | -46 | 1.24 | 0.22 |
| Glucose (mg/dl) | 90±13 | 88±13 | -2.4 | 1.2 | 0.23 |
| HDL-cholesterol (mg/dl) | 51±12 | 46±13 | -12 | 3.1 | <0.0033 |
| Non-HDL-cholesterol (mg/dl) | 142±41 | 134±29 | -7.8 | 1.8 | 0.07 |
| Triglycerides (mg/dl) | 186±218 | 138±83 | -48 | 2.1 | <0.039 |
| Insulin (μUI/ml) | 15.2±9.1 | 13.1±9.0 | -2.1 | 2.6 | <0.011 |
| HOMA-IR | 3.5±2.2 | 2.9±2.1 | -0.6 | 3.2 | <0.0024 |
| Ghrelin (μg/ml) | 84.8±43.3 | 81.7±45.8 | -3.1 | 0.9 | 0.38 |
| Leptin (pg/ml) | 43.5±24.9 | 41.1±24.2 | -2.4 | 0.9 | 0.35 |
| Group 2 Innate lymphoid cells (ILC2) (%) | 2.7±2.5 | 5.8±4.5 | 2.8 | -2.9 | <0.008 |
| Non-classical monocytes (CD14$^-$CD16$^{++}$) (%) | 7.2±6.1 | 5.3±3.7 | -1.8 | 2.4 | <0.02 |
| Intermediate monocytes (CD14$^+$CD16$^+$) (%) | 8.0±4.7 | 7.5±5.5 | -0.5 | 0.5 | 0.61 |
| Classical Monocytes (CD14$^+$CD16$^-$) (%) | 62.9±15.3 | 71.0±10.6 | 6.7 | -3.3 | <0.0015 |
| CD16$^+$/CD16$^-$ ratio | 0.3±0.2 | 0.2±0.1 | -0.08 | 2.5 | <0.015 |
| β2AR expression on NCM (MFI) | 10778±5489 | 10501±6180 | -1743 | 0.25 | 0.80 |
| β2AR expression on IM (MFI) | 27069±15070 | 23350±13916 | -6480 | 1.7 | <0.05 |
| β2AR expression on CM (MFI) | 14343±8117 | 13626±8496 | -2785 | 0.5 | 0.60 |

n = 52 obese subjects. Differences between groups were evaluated with Student's T test. $p < 0.05$ was considered statistically significant. REE, resting energy expenditure CR, caloric restriction; BMI, body mass index; HOMA-IR, homeostatic model assessment- insulin resistance; EE, energy expenditure; β2AR, beta-2 adrenergic receptor; NCM, non-classical monocytes (CD14$^-$CD16$^{++}$); IM, intermediate monocytes (CD14$^+$CD16$^+$); CM, classical monocytes (CD14$^+$CD16$^+$); MFI, mean fluorescence intensity.

perturbations of immune system and dysregulated adipose tissue (AT) homeostasis [30]. Our findings, using a comprehensive multiparameter cytometry analysis of Mo and ILC2s in peripheral blood, provide further support to their role in metabolic homeostasis and the physiopathology of AT.

We found circulating NCMs increased in the obese group, but CMs and IMs did not change. Previous studies on the abundance of circulating CM (CD14$^+$CD16$^-$) in obesity are inconsistent, with reports showing decrease or no change [15]. Ours results, support the concept of increased circulating CD16$^+$ monocyte subpopulations (IM and NCM) in obesity associated with cardio-metabolic risk factors [15–20]. The intermediate subtype (CD14$^+$CD16$^+$) is a small percentage of transitional monocytes with phagocytic and more pro-inflammatory capacity than the NCM [15,31,32]. Zawada *et al* [33] proposed a pro-angiogenic behaviour for IM, suggesting that the increase in peripheral blood has an important role for inflammation and progression into atherosclerosis [32]. In contrast to Krinninger *et al* [32], who found increase the percentage of CD14$^+$CD16$^+$ monocytes, we only found that IM count had a non-significant trend to increase. Despite these results, it is possible that in obesity occurs a shift of monocytes toward a pro-inflammatory phenotype.

Innate lymphoid type 2 cells (ILC2s) are key regulators of the immune and metabolic homeostasis of visceral adipose tissue (VAT) and may be determinants of weight, considering their involvement in beige fat development, through IL-5 [34–37]. Altered ILC2s amounts and function have been found in VAT but not in circulation in humans and other species with

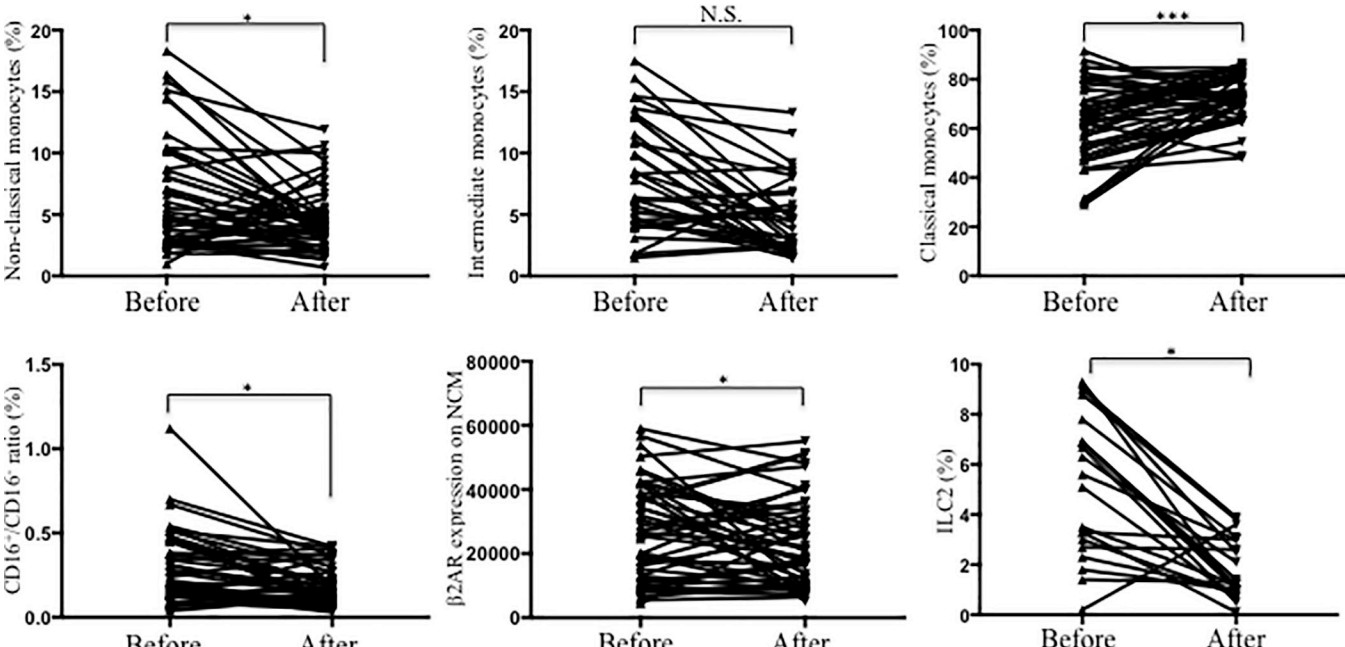

**Fig 5. Changes in Mo subsets, circulating ILC2s and β2AR expression after diet.** Graphs of symbols & lines from obese subjects are shown before and after intervention for **a-d**) Percentage of Mo subsets, **e**) β2AR expression on IM, and **f**) percentage of ILC2s. *$p < 0.05$, **$p < 0.01$, ***$p < 0.001$.

metabolic disorders such as obesity. In the present work, we developed an assay to assess circulating ILC2 by four-color flow cytometry following directions of previous scientific works in allergy [38–40]. We demonstrated that obese subjects have decreased circulating ILC2s. To our knowledge, this is the first report using this procedure; therefore we cannot compare our results with other investigations. Multiparameter flow cytometry of freshly drawn peripheral blood offers the promise of a highly sensitive and reproducible approach, which allows the identification and quantification of complex cell subpopulations, such as ILCs.

Obesity is also linked to altered hypothalamic–pituitary–adrenal axis (HPA) and sympathetic nervous system (SNS) function, triggering inflammation which increases β2AR expression in peripheral blood mononuclear cells [41]. In our study, intermediate monocytes (CD14+CD16+) from obese individuals expressed more β2AR. Our results agree with those of Gálvez *et al* [42] who described β2AR induction of a shift towards an anti-inflammatory phenotype profile. In addition, Hong *et al* [43] demonstrated higher amounts of β2AR on monocytes with reduced responsiveness. Due to the role of sympathetic activation in hypertension and cardiovascular pathology is well studied, and the IM are involved in cardiovascular events, the overexpression of β2AR is in agreement with the increased frequency of hypertension and cardiovascular damage in obese patients. Therefore, sympatho-adrenal regulation in monocytes is an important aspect of vascular inflammation. In addition, β2AR may act as a molecular rheostat to fine-tune anti-inflammatory responses preventing inflammation [44].

Inflammation modifies energy metabolism, enhancing energy expenditure, and reducing energy intake, and induces AT remodelling [2]. The associations of caloric intake, positive with IM subset, and negative with ILC2s, support the pro-inflammatory environment of the metabolic imbalance and loss of AT browning observed in obesity.

Ghrelin and leptin are important components of the neuroendocrine control of energy homeostasis and immune system regulation [9,10,45,46]. In our work, acylated-ghrelin was associated negatively with NCM, which we explain by the ghrelin action as an anti-

**Table 4. Associations of changes in monocytes subsets, ILC2s, and β2AR expression within Mo subsets with anthropometric and metabolic features after dietary restriction.**

| Dependent variable | Regressors | β±S.E. | T | p-level |
|---|---|---|---|---|
| $\Delta$**Non-Classical monocytes** (NCM; CD14-CD16++). Adjusted $R^2$ = 0.94 | | | | |
| | Intercept | | -4.07 | <0.00013 |
| | $\Delta$Leptin levels | -0.27±0.12 | -2.22 | <0.03 |
| | $\Delta$HDL-C | -0.24±0.12 | -1.96 | <0.046 |
| $\Delta$**Intermediate monocytes** (IM; CD14+CD16+) Adjusted $R^2$ = 0.134 | | | | |
| | Intercept | | 2.25 | <0.04 |
| | $\Delta$Leptin levels | -0.34±0.12 | -2.91 | <0.01 |
| | $\Delta$HDL-C | -0,24±0.12 | -2.02 | <0.047 |
| $\Delta$**Classical monocytes** (CM; CD14+CD16-) Adjusted $R^2$ = 0.157 | | | | |
| | Intercept | | 3.79 | <0.0001 |
| | $\Delta$HDL-cholesterol | 0.37±0.11 | 3.20 | <0.002 |
| $\Delta$**CD16+/CD16- ratio** Adjusted $R^2$ = 0.316 | | | | |
| | Intercept | | -4.49 | <0.00003 |
| | $\Delta$HDL-Cholesterol | -0.34±0.10 | -3.29 | <0.0017 |
| | $\Delta$Leptin | -0.31±0.11 | -2.82 | <0.0065 |
| | $\Delta$BMI | -0.32±0.11 | -2.82 | <0.0065 |
| | $\Delta$REE | 0.23±0.11 | 2.08 | <0.04 |
| $\Delta$**Group 2 innate lymphoid cells (ILC2s)** Adjusted $R^2$ = 0.143 | | | | |
| | Intercept | | 7.37 | <0.0000001 |
| | $\Delta$HOMA-IR | -0.40±0.12 | -3.42 | <0.0011 |
| $\Delta$**β2AR expression on NCM** Adjusted $R^2$ = 0.401 | | | | |
| | Intercept | | -5.04 | <0.000005 |
| | $\Delta$BMI | -0.53±0.10 | -5.12 | <0.000003 |
| | $\Delta$Leptin | 0.49±0.11 | 4.66 | <0.000018 |
| | $\Delta$HDL-C | 0.25±0.10 | 2.63 | <0.011 |
| | $\Delta$HOMA-IR | -0.24±0.10 | -2.37 | <0.021 |
| $\Delta$**β2AR expression on IM** Adjusted $R^2$ = 0.058 | | | | |
| | Intercept | | -3.09 | <0.0030 |
| | $\Delta$BMI | -0.27±0.12 | -2.22 | <0.029 |
| $\Delta$**β2AR expression on CM** Adjusted $R^2$ = 0.258 | | | | |
| | Intercept | | -4.54 | <0.000026 |
| | $\Delta$BMI | -0.53±0.11 | -4.65 | <0.000018 |
| | $\Delta$Leptin | 0.25±0.11 | 2.19 | <0.032 |

The associations were evaluate by multiple regression (n = 52). $p<0.05$ was considered statistically significant.

$\Delta$, delta; BMI, body mass index; REE, energy expenditure; β2AR, beta-2 adrenergic receptor; NCM, non-classical monocytes (CD14-CD16++); IM, intermediate monocytes (CD14+CD16+); CM, classical monocytes (CD14+CD16+).

inflammatory cytokine in homeostasis with potent orexigenic effect. Total ghrelin levels are reduced in obese patients. In addition, a lack of ghrelin signalling or increase conversion to the desacyl form may exacerbate the inflammatory response [10,46]. In contrast, leptin, a pro-inflammatory cytokine, has an important role in the control of energy metabolism and the metabolism-immune interplay [47]. Leptin promotes the proliferation and activation of NCM and IM favouring metabolic diseases, such as obesity [45,48], thereby leptin action induces higher BMI and REE as an effect of the chronic low-grade inflammation *per se*, which could explain the increase of β2AR expression on monocytes and its association with BMI.

We found a positive relationship between insulin-resistance and NCMs, suggesting that obesity-induced insulin resistance aggravates the chronic low-grade inflammation favoring the shift to increase the CD14⁻CD16⁺⁺ subset (NCM) [21,49]. We also confirmed the relations of NCMs and ILC2s with plasma lipids. Increased CD14⁺CD16⁺⁺ monocytes and low HDL-C levels are reported in inflammatory disorders [22]. HDL-C and Apo A-1 may prevent monocytes activation and their attachment to the endothelium surface [50]. Furthermore, HDL-cholesterol levels are positively associated with ILC2s. This suggests that HDL-C are involved in cardiovascular protection due to its anti-inflammatory, antioxidant, and antithrombotic properties [35,51]. Furthermore, dietary intervention reverses HDL-C levels and ILC2s.

Since dietary habits influence energy and immune homeostasis, weight loss in obesity has a beneficial effect in the immune system, mediated by secretion of hormones and cytokines [52]. In this work, the inflammatory status decreased under caloric restriction mediated probably by the increase of CM and ILC2s. Few studies explore the impact of different types of diets on weight loss, the immune response [53] and immune cells [54,55]. We found that short-term caloric restriction induced a 4.5% weight loss diminishing CD14⁻CD16⁺⁺ (NCM) and CD14⁺CD16⁺ (IM) monocyte numbers, pointing to a decrease of chronic low-grade inflammation. In contrast, Manco *et al* [56] reported that a 5% weight loss do not induce an effect on systemic or subcutaneous adipose tissue markers of inflammation. However, our findings agree with other works *e.g.* Nieman *et al* [57] mentioned that a moderate weight loss, decrease certain aspects of immune system (percentage of immune cells and functions); and Kim *et al* demonstrated that caloric restriction with a high protein diet, also decreases NCM subset [22]. Considering that we administered a diet with moderate caloric restriction we concluded that this diet might offer the benefit of reduction of inflammatory damage, even with moderate weight loss.

In our work, circulating ILC2s augmented after dietary intervention, which led a decrease of REE by the mediation of IL-5 secretion [26,51], therefore, ILC2s might balance visceral metabolic homeostasis and promote beige cells expansion, which regulates REE. The findings of Brestoff et al. [26] also support the idea that ILC2s may be a key immune component of the thermogenic circuit and a determinant of adipose tissue metabolic status.

The decreased β2AR expression on IM after caloric restriction may contribute to mitigate inflammation. We found that after caloric restriction the changes of β2AR expression on three subpopulations of monocytes correlated with diminution of BMI, in agreement with previous reports [42,57].

## Conclusions

This study shows several novel findings: the increase of NCM, decreases of circulating ILC2s, and higher β2AR expression on IM in obese subjects, the positive associations between caloric intake and IM, and negative with ILC2s, gives further support to the concept of the participation of inflammation in energy expenditure. We observed after intervention a decrease of circulating NCM, and an increase of CM, and ILC2s. In conclusion, our work underscores the link between obesity and pro-inflammatory environmental that influences impairments of immune system and metabolism, responses that may be modified after dietary intervention.

## Supporting information

**S1 Table.**
(XLSX)

## Author Contributions

**Conceptualization:** Nicté Figueroa-Vega, Juan Manuel Malacara.

**Formal analysis:** Nicté Figueroa-Vega.

**Funding acquisition:** Juan Manuel Malacara.

**Investigation:** Nicté Figueroa-Vega.

**Methodology:** Nicté Figueroa-Vega, Carolina I. Marín-Aragón, Itzel López-Aguilar, Lorena Ibarra-Reynoso, Elva Pérez-Luque.

**Resources:** Juan Manuel Malacara.

**Validation:** Nicté Figueroa-Vega.

**Writing – original draft:** Nicté Figueroa-Vega, Elva Pérez-Luque, Juan Manuel Malacara.

**Writing – review & editing:** Nicté Figueroa-Vega, Elva Pérez-Luque, Juan Manuel Malacara.

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
