## [Decision Letter · Decision Letter 0]

24 Oct 2019

PONE-D-19-23742

Analysis of the percentages of monocyte subsets and ILC2s, their relationships with metabolic variables and response to hypocaloric restriction in obesity

PLOS ONE

Dear Dr Malacara,

Thank you for submitting your manuscript to PLOS ONE. After careful consideration, we feel that it has merit but does not fully meet PLOS ONE’s publication criteria as it currently stands. Therefore, we invite you to submit a revised version of the manuscript that addresses the points raised during the review process.

PLOS ONE's publication criteria can be found here: https://journals.plos.org/plosone/s/criteria-for-publication

In particular, please make sure your manuscript meets point 5 ("The article is presented in an intelligible fashion and is written in standard English"), as PLOS ONE does not use a copy editor and there are numerous small errors throughout the manuscript. Point 3 (Experiments, statistics, and other analyses are performed to a high technical standard and are described in sufficient detail) also needs to be met.

We would appreciate receiving your revised manuscript by Dec 08 2019 11:59PM. To enhance the reproducibility of your results, we recommend that if applicable you deposit your laboratory protocols in protocols.io, where a protocol can be assigned its own identifier (DOI) such that it can be cited independently in the future. For instructions see: http://journals.plos.org/plosone/s/submission-guidelines#loc-laboratory-protocols

We look forward to receiving your revised manuscript.

Kind regards,

Melissa M Markofski

Academic Editor

PLOS ONE

Journal Requirements:

'This work was supported by the grant CB2014-242065M from Consejo Nacional de Ciencia y Tecnología

(CONACYT Ciencia Básica 2014 México, to JMM).'

'The funders had no role in study design, data collections and analysis, decision to

publish or preparation of the manuscript'

Additional Editor Comments:

In addition to the points made by the reviewers, please add more details for your flow cytometery methodology. How did you set your gates? For example, were FMOs only used for ICL2s? The gating strategy needs more detail, and explaining the methods will help strengthen the manuscript. 

Why are all your cells in Figure 2 positive? It seems there should be cells that are not β2AR positive. 

In addition, your monocyte gate is quite small, which is particularly worrisome since the voltage is such that there is not much “separation” between cell populations. (for future research studies, you may want to consider adjusting your voltages so that it is easier to distinguish between cell populations) 

Please carefully go through your results and discussion sections and make the results more specific. As it is written right now, it is unclear which results are being compared. For example “No differences in β2AR expression by NCM (CD14-CD16++) and CM (CD14++CD16-) were observed between the groups of obese and non-obese subjects. Yet, the IM (CD14+CD16+) from obese subjects expressed significantly more β2AR (p<0.001) (Table 1 and Fig. 2).” These sections are missing which time points are being compared. This is a consistent problem throughout the results and discussion sections.

Much of the discussion session needs to be re-written. The current discussion section is written as a literature review, as there is very little discussion of the results of your study and your interpretation of what these findings mean. Although the discussion section generally avoids over-stating, it does need more work to discuss the findings of the research study. 

Reviewers' comments:

Reviewer's Responses to Questions

**Comments to the Author**

1. Is the manuscript technically sound, and do the data support the conclusions?

Reviewer #1: Partly

Reviewer #2: Yes

2. Has the statistical analysis been performed appropriately and rigorously? 

Reviewer #1: Yes

Reviewer #2: Yes

3. Have the authors made all data underlying the findings in their manuscript fully available?

Reviewer #1: Yes

Reviewer #2: Yes

4. Is the manuscript presented in an intelligible fashion and written in standard English?

Reviewer #1: Yes

Reviewer #2: No

5. Review Comments to the Author

Reviewer #1: I appreciate the opportunity to review this interesting paper on circulating monocytes and group 2 innate lymphoid cells (ILC2s) in lean and obese (obesity assessed by BMI) subjects, and the possible effect of hypocaloric restriction on these cells. The premise of this study is the hypothesis that in obese subjects non-classical monocytes subpopulation is increased and ILC2s are decreased comparing to lean controls and that the weight loss induced by hypocaloric restriction hypocaloric would reverse the situation. Authors has tested this hypothesis on 74 lean and 65 obese individuals, who were subject to diet restrictions for 7-weeks. They observed that In obese individuals the percent of non-classical monocytes (NCM), and the expression of the �2-adrenergic receptor (�2AR) expression by intermediate monocytes (IM) were increased, whereas the percent of ILC2s was decreased. Caloric restriction lead to a decrease of NCM and the �2AR expression by IM, as well as, an increase in the percent of classical monocytes and ILC2s. Authors concluded that the weight loss induced by hypocaloric restriction is significantly associated with changes of monocytes and ILC2s percentages, which may contribute to the attenuation of the low-grade inflammation linked to obesity.

The main strengths of this paper is that it addresses an interesting and timely problem, the article is well constructed, the experiments were well conducted, and analysis was well performed. Recently, ILC2s were identified in murine and human adipose tissue and have been shown to promote the white adipose tissue beiging and prevent the development of obesity

Considering these strengths, I feel some concerns on the manuscript, which should be addressed before being accepted for the publication.

Major comments:

• The research question is not clear from the introduction. Authors should more convincingly justify why they decided to measure subpopulations of monocytes and ILC2. Why do the authors think that it is important to measure circulating ILC2s in the blood when their action appears to be mainly local, in adipose tissue? Please discuss. Authors should also justify in the introduction why decide to measure leptin and ghrelin.

• Reliance on BMI as a sole marker of obesity, seems to be the serious limitation of the study. A number of authors indicated a poor linear relationship between BMI and total body fat and also suggested that body fat distribution would be more clinically significant than overall obesity. For instance using dual energy x-ray absorptiometry (DXA) seems to be fast and relatively inexpensive method to assess visceral adipose tissue.

• A flow chart showing the experimental procedure should be included as it makes the experimental setting more visible to the reader. It isn’t clear for me what were the effects of caloric restriction in non-obese participants?

• Conclusions are too general.

Minor comments:

• How exactly participants were selected for the study? I assume that individuals with BMI ≥24.9kg/m2 and ≤30 (kg/m2 were excluded from the study. Authors should clearly state it. In parasite infections and allergies, circulating ILC2s could be elevated. How did authors ensure exclusion such participants?

• Figures are confusing and the figure legends do not provide sufficient information to describe the data.

Reviewer #2: The purpose of the current study was to assess monocyte subsets and ILC2s in lean and obese subjects, and the possible

effect of energy restriction on these innate cells. The authors found that the percentage of non-classical monocytes and the expression of the Beta-2AR by intermediate monocytes were increased. In contrast the the percentage of ILC2s

was decreased in subjects with obesity. The authors also showed that there were significant negative associations between

ILC2 and caloric intake, and Beta-2AR expression by intermediate monocytes and resting energy expenditure, but a positive relationship between non-classical monocytes and insulin resistance. Caloric restriction reduced non-classical monocytes, and Beta-2AR expression by intermediate monocytes. Classical monocyte and ILC2 populations increased with caloric restriction as well. These results suggest that weight loss may contribute to the reduction in low-grade inflammation in part by increasing ILC2 cell numbers and modifying the inflammatory characteristics of circulating monocytes. Overall, these results are novel and interesting but there were numerous grammatical errors (too many to list) throughout the manuscript that should be corrected.

6. PLOS authors have the option to publish the peer review history of their article (what does this mean?). If published, this will include your full peer review and any attached files.

Reviewer #1: No

Reviewer #2: No

---

## [Author Response · Author response to Decision Letter 0]

22 Nov 2019

Additional Editor Comments:

In addition to the points made by the reviewers, please add more details for your flow cytometery methodology. How did you set your gates? For example, were FMOs only used for ICL2s? The gating strategy needs more detail, and explaining the methods will help strengthen the manuscript. 

R = We included this information in the new version of manuscript, (M&M, results and figure legends sections) (page 6 lines 129-134, and 140-144; page 7 lines 145-148 and 156-161).

Why are all your cells in Figure 2 positive? It seems there should be cells that are not β2AR positive.

R = We used unstained cells tube and isotype control tube in each experiment for assigning the gates and markers for negative and positive fluorescence. 

In addition, your monocyte gate is quite small, which is particularly worrisome since the voltage is such that there is not much “separation” between cell populations. (for future research studies, you may want to consider adjusting your voltages so that it is easier to distinguish between cell populations).

R = We re-edited the dot plots and gates with Kaluza software included the total events acquired (50,000). These figures were included in the new version of Figure 1 and figure 2.

Please carefully go through your results and discussion sections and make the results more specific. As it is written right now, it is unclear which results are being compared. For example “No differences in β2AR expression by NCM (CD14-CD16++) and CM (CD14++CD16-) were observed between the groups of obese and non-obese subjects. Yet, the IM (CD14+CD16+) from obese subjects expressed significantly more β2AR (p<0.001) (Table 1 and Fig. 2).” These sections are missing which time points are being compared. This is a consistent problem throughout the results and discussion sections.

R = We apologized by unclear redaction. We changed this paragraph in the new version of manuscript (page 9 lines 190-192).

Much of the discussion session needs to be re-written. The current discussion section is written as a literature review, as there is very little discussion of the results of your study and your interpretation of what these findings mean. Although the discussion section generally avoids over-stating, it does need more work to discuss the findings of the research study. 

R = According to this observation, we included in the new version of manuscript a discussion more detailed about our findings.

Reviewers' comments:

Reviewer's Responses to Questions

Comments to the Author

1. Is the manuscript technically sound, and do the data support the conclusions?

Reviewer #1: Partly 

R = According to this observation, we included in the new version of manuscript a discussion more detailed about our findings.

Reviewer #2: Yes

2. Has the statistical analysis been performed appropriately and rigorously? 

Reviewer #1: Yes

Reviewer #2: Yes

3. Have the authors made all data underlying the findings in their manuscript fully available?

Reviewer #1: Yes

Reviewer #2: Yes

4. Is the manuscript presented in an intelligible fashion and written in standard English?

Reviewer #1: Yes

Reviewer #2: No

R = We reviewed the typographical and grammatical mistakes and we offer you a new corrected version.

5. Review Comments to the Author

Reviewer #1: I appreciate the opportunity to review this interesting paper on circulating monocytes and group 2 innate lymphoid cells (ILC2s) in lean and obese (obesity assessed by BMI) subjects, and the possible effect of hypocaloric restriction on these cells. The premise of this study is the hypothesis that in obese subjects non-classical monocytes subpopulation is increased and ILC2s are decreased comparing to lean controls and that the weight loss induced by hypocaloric restriction hypocaloric would reverse the situation. Authors has tested this hypothesis on 74 lean and 65 obese individuals, who were subject to diet restrictions for 7-weeks. They observed that In obese individuals the percent of non-classical monocytes (NCM), and the expression of the β2-adrenergic receptor (β2AR) expression by intermediate monocytes (IM) were increased, whereas the percent of ILC2s was decreased. Caloric restriction lead to a decrease of NCM and the β2AR expression by IM, as well as, an increase in the percent of classical monocytes and ILC2s. Authors concluded that the weight loss induced by hypocaloric restriction is significantly associated with changes of monocytes and ILC2s percentages, which may contribute to the attenuation of the low-grade inflammation linked to obesity.

The main strengths of this paper is that it addresses an interesting and timely problem, the article is well constructed, the experiments were well conducted, and analysis was well performed. Recently, ILC2s were identified in murine and human adipose tissue and have been shown to promote the white adipose tissue beiging and prevent the development of obesity

Considering these strengths, I feel some concerns on the manuscript, which should be addressed before being accepted for the publication.

Major comments:

• The research question is not clear from the introduction. Authors should more convincingly justify why they decided to measure subpopulations of monocytes and ILC2. Why do the authors think that it is important to measure circulating ILC2s in the blood when their action appears to be mainly local, in adipose tissue? Please discuss. Authors should also justify in the introduction why decide to measure leptin and ghrelin.

R = Both explanation were added in Introduction and Discussion sections (page 4, lines 78-80, and page 12 lines 239-242).

• Reliance on BMI as a sole marker of obesity, seems to be the serious limitation of the study. A number of authors indicated a poor linear relationship between BMI and total body fat and also suggested that body fat distribution would be more clinically significant than overall obesity. For instance using dual energy x-ray absorptiometry (DXA) seems to be fast and relatively inexpensive method to assess visceral adipose tissue.

R = We agree with the reviewer contention that BMI has a poor linear relationship with total body mass. However, we do not agree with the affirmation that it is a serious limitation for the diagnosis of obesity. Current criteria widely used for the diagnosis of obesity, is based on BMI and not on the assessment of visceral body fat with DXA.

• A flow chart showing the experimental procedure should be included as it makes the experimental setting more visible to the reader. It isn’t clear for me what were the effects of caloric restriction in non-obese participants?

R = We included at final of Ms this flow chart which explains the recruitment of subjects and procedures. Diet was prescribed only for obese individuals as stated in M&M, Normal weight subjects continued with their customary diet.

• Conclusions are too general.

R = We offer you a new version of conclusions.

Minor comments:

• How exactly participants were selected for the study? I assume that individuals with BMI ≥24.9kg/m2 and ≤30 (kg/m2 were excluded from the study. Authors should clearly state it. In parasite infections and allergies, circulating ILC2s could be elevated. How did authors ensure exclusion such participants?

R = We explained in more detail the selection in participants section and in the flow chart added at the end of M&M section. Subjects with overweight were not included in the study (≥24.9 kg/m2 and ≤30 kg/m2). Only recruitment individuals with >30BMI<35 kg/m2). In addition, as stated in M&M, we did not included subjects with infections (allergies, flu, cold, dental diseases, etc).

• Figures are confusing and the figure legends do not provide sufficient information to describe the data.

R = We re-edited the figures and M&M and figure legends sections in the new version of manuscript (page 6, lines 129-134 and 139-144; page 7, lines 145-149, and 150-161; page 32, lines 524-528, 532-535 and 539-548). 

Reviewer #2.

The purpose of the current study was to assess monocyte subsets and ILC2s in lean and obese subjects, and the possible effect of energy restriction on these innate cells. The authors found that the percentage of non-classical monocytes and the expression of the Beta-2AR by intermediate monocytes were increased. In contrast the percentage of ILC2s was decreased in subjects with obesity. The authors also showed that there were significant negative associations between ILC2 and caloric intake, and Beta-2AR expression by intermediate monocytes and resting energy expenditure, but a positive relationship between non-classical monocytes and insulin resistance. Caloric restriction reduced non-classical monocytes, and Beta-2AR expression by intermediate monocytes. Classical monocyte and ILC2 populations increased with caloric restriction as well. These results suggest that weight loss may contribute to the reduction in low-grade inflammation in part by increasing ILC2 cell numbers and modifying the inflammatory characteristics of circulating monocytes. Overall, these results are novel and interesting but there were numerous grammatical errors (too many to list) throughout the manuscript that should be corrected.

 R = We offer you a new corrected version of manuscript ________________________________________

6. PLOS authors have the option to publish the peer review history of their article (what does this mean?). If published, this will include your full peer review and any attached files.

Do you want your identity to be public for this peer review? For information about this choice, including consent withdrawal, please see our Privacy Policy.

Reviewer #1: No

Reviewer #2: No

---

## [Editor Report · Decision Letter 1]

26 Nov 2019

PONE-D-19-23742R1

Analysis of the percentages of monocyte subsets and ILC2s, their relationships with metabolic variables and response to hypocaloric restriction in obesity

PLOS ONE

Dear Dr Malacara,

Thank you for submitting your manuscript to PLOS ONE. After careful consideration, we feel that it has merit but does not fully meet PLOS ONE’s publication criteria as it currently stands. Therefore, we invite you to submit a revised version of the manuscript that addresses the points raised during the review process.

I have received your revised manuscript, however there are a few things that need to be addressed before it is sent out for peer review. Figures are missing from the revised manuscript (only the flow chart is included in the revised manuscript, which is not given a figure number). In addition, in the authors' response it is stated that the "Increased β2AR expression by intermediate monocytes" paragraph has been revised. However, it is still exactly the same wording and still does not tell the reader what time points are different. Please go through the responses to make sure that all revisions that stated were completed were actually completed in the manuscript.  

We would appreciate receiving your revised manuscript by Jan 10 2020 11:59PM. To enhance the reproducibility of your results, we recommend that if applicable you deposit your laboratory protocols in protocols.io, where a protocol can be assigned its own identifier (DOI) such that it can be cited independently in the future. For instructions see: http://journals.plos.org/plosone/s/submission-guidelines#loc-laboratory-protocols

We look forward to receiving your revised manuscript.

Kind regards,

Melissa M Markofski

Academic Editor

PLOS ONE

---

## [Author Response · Author response to Decision Letter 1]

10 Jan 2020

January 10th, 2019

Additional Editor Comments:

In addition to the points made by the reviewers, please add more details for your flow cytometry methodology. How did you set your gates? For example, were FMOs only used for ICL2s? The gating strategy needs more detail, and explaining the methods will help strengthen the manuscript. 

R = According to this observation, we have included this information in the corrected manuscript, in the Materials and Methods section (page 6, lines 130-135, and 139-143; page 7, lines 144-146 and 147-158), and figure legends (page 31, lines 513-516, 520-524, and 531-536). 

Why are all your cells in Figure 2 positive? It seems there should be cells that are not β2AR positive.

R = According to this comment, we have added a new histogram of β2AR staining showing the percents of positive and negative cells, according to a negative (cells stained with an irrelevant isotype-matched mAb. In addition, and as stated in the corrected manuscript, results have been expressed as the MFI of positive cells. We used unstained cells tube and isotype controls tube in each experiment for assigning the gates and markers for negative and positive fluorescence. In the new Figure 2 we now included a histogram of PE-isotype control with its marker corresponding. Also in Figure Legend 2 included this information (page 31, lines 527-536).

In addition, your monocyte gate is quite small, which is particularly worrisome since the voltage is such that there is not much “separation” between cell populations. (for future research studies, you may want to consider adjusting your voltages so that it is easier to distinguish between cell populations).

R = We fully appreciate this recommendation. According to it, the dot plots and gates have been corrected, by using the Kaluza software. These figures have been included in the new version of the manuscript (Figs. 1 and 2).

Please carefully go through your results and discussion sections and make the results more specific. As it is written right now, it is unclear which results are being compared. For example “No differences in β2AR expression by NCM (CD14-CD16++) and CM (CD14++CD16-) were observed between the groups of obese and non-obese subjects. Yet, the IM (CD14+CD16+) from obese subjects expressed significantly more β2AR (p<0.001) (Table 1 and Fig. 2).” These sections are missing which time points are being compared. This is a consistent problem throughout the results and discussion sections.

R = We apologize by the confuse and improper writing of these paragraphs. According to this observation, these paragraphs have been corrected (page 8, lines 187-189).

Much of the discussion session needs to be re-written. The current discussion section is written as a literature review, as there is very little discussion of the results of your study and your interpretation of what these findings mean. Although the discussion section generally avoids over-stating, it does need more work to discuss the findings of the research study. 

R = According to this observation, the Discussion section has been thoroughly revised, with a proper analysis of our findings.

Reviewers' comments:

Reviewer's Responses to Questions

Comments to the Author

1. Is the manuscript technically sound, and do the data support the conclusions?

Reviewer #1: Partly 

R = According to this observation, we included in the new version of manuscript a discussion more detailed about our findings.

Reviewer #2: Yes

2. Has the statistical analysis been performed appropriately and rigorously? 

Reviewer #1: Yes

Reviewer #2: Yes

3. Have the authors made all data underlying the findings in their manuscript fully available?

Reviewer #1: Yes

Reviewer #2: Yes

4. Is the manuscript presented in an intelligible fashion and written in standard English?

Reviewer #1: Yes

Reviewer #2: No

R = We have reviewed the grammar and syntax of the manuscript.

5. Review Comments to the Author

Reviewer #1: I appreciate the opportunity to review this interesting paper on circulating monocytes and group 2 innate lymphoid cells (ILC2s) in lean and obese (obesity assessed by BMI) subjects, and the possible effect of hypocaloric restriction on these cells. The premise of this study is the hypothesis that in obese subjects non-classical monocytes subpopulation is increased and ILC2s are decreased comparing to lean controls and that the weight loss induced by hypocaloric restriction hypocaloric would reverse the situation. Authors has tested this hypothesis on 74 lean and 65 obese individuals, who were subject to diet restrictions for 7-weeks. They observed that In obese individuals the percent of non-classical monocytes (NCM), and the expression of the β2-adrenergic receptor (β2AR) expression by intermediate monocytes (IM) were increased, whereas the percent of ILC2s was decreased. Caloric restriction lead to a decrease of NCM and the β2AR expression by IM, as well as, an increase in the percent of classical monocytes and ILC2s. Authors concluded that the weight loss induced by hypocaloric restriction is significantly associated with changes of monocytes and ILC2s percentages, which may contribute to the attenuation of the low-grade inflammation linked to obesity.

The main strengths of this paper is that it addresses an interesting and timely problem, the article is well constructed, the experiments were well conducted, and analysis was well performed. Recently, ILC2s were identified in murine and human adipose tissue and have been shown to promote the white adipose tissue beiging and prevent the development of obesity

Considering these strengths, I feel some concerns on the manuscript, which should be addressed before being accepted for the publication.

Major comments:

• The research question is not clear from the introduction. Authors should more convincingly justify why they decided to measure subpopulations of monocytes and ILC2. Why do the authors think that it is important to measure circulating ILC2s in the blood when their action appears to be mainly local, in adipose tissue? Please discuss. Authors should also justify in the introduction why decide to measure leptin and ghrelin.

R = According to this observation, we have added relevant information regarding adipokines (page 3, lines 51-58), and immune cells (page 4 lines 66-69, 78-80, and 86-88) in the Introduction as well as in the Discussion (page 11, lines 238-240, and 255-259; and page 12, lines 260-261). We hope that this reviewer detects now more convincing information in the corrected manuscript that justifies the analysis of the different immune and metabolic parameters detected in our study. 

• Reliance on BMI as a sole marker of obesity, seems to be the serious limitation of the study. A number of authors indicated a poor linear relationship between BMI and total body fat and also suggested that body fat distribution would be more clinically significant than overall obesity. For instance using dual energy x-ray absorptiometry (DXA) seems to be fast and relatively inexpensive method to assess visceral adipose tissue.

R = We agree that BMI has a poor linear relationship with total body fat. However, we consider that BMI is still a useful parameter for the diagnosis of obesity. In addition, unfortunately, in our University we do not have access to DXA.

• A flow chart showing the experimental procedure should be included as it makes the experimental setting more visible to the reader. It isn’t clear for me what were the effects of caloric restriction in non-obese participants?

R = According to this observation, a flow chart accounting for the recruitment of subjects and experimental procedures has been included in the corrected manuscript (page 33). Moreover, the caloric restriction diet was prescribed only for obese individuals, as stated in the Materials and Methods section of the corrected manuscript (page 5, lines 100-101, and 111-112, and page 6, lines 119-120).

• Conclusions are too general.

R = We are offering a new version of conclusions, with a proper analysis of our findings.

Minor comments:

• How exactly participants were selected for the study? I assume that individuals with BMI ≥24.9kg/m2 and ≤30 (kg/m2 were excluded from the study. Authors should clearly state it. In parasite infections and allergies, circulating ILC2s could be elevated. How did authors ensure exclusion such participants?

R = According to this observation, a more detailed description of the inclusion criteria for the recruitment of individuals has been added to the corrected manuscript, including the new flow chart (page 33). Moreover, it is worth mentioning that subjects with overweight were not included in the study (≥24.9 kg/m2 and ≤30 kg/m2), and that those subjects with atopy, allergic symptoms or with parasite infections (according to their medical records) were also excluded from the study. 

• Figures are confusing and the figure legends do not provide sufficient information to describe the data.

R = We re-edited the figures 1, 2 and 3, called now as New Fig. 1, New Fig. 2 and New Fig. 3. The information was added in M&M (page 6, lines 130-135, and 139-143; page 7, lines 146-147 and 150-159), and figure legends (page 312, lines 514-517, 520-525, and 528-537). 

Reviewer #2.

The purpose of the current study was to assess monocyte subsets and ILC2s in lean and obese subjects, and the possible effect of energy restriction on these innate cells. The authors found that the percentage of non-classical monocytes and the expression of the Beta-2AR by intermediate monocytes were increased. In contrast the percentage of ILC2s was decreased in subjects with obesity. The authors also showed that there were significant negative associations between ILC2 and caloric intake, and Beta-2AR expression by intermediate monocytes and resting energy expenditure, but a positive relationship between non-classical monocytes and insulin resistance. Caloric restriction reduced non-classical monocytes, and Beta-2AR expression by intermediate monocytes. Classical monocyte and ILC2 populations increased with caloric restriction as well. These results suggest that weight loss may contribute to the reduction in low-grade inflammation in part by increasing ILC2 cell numbers and modifying the inflammatory characteristics of circulating monocytes. Overall, these results are novel and interesting but there were numerous grammatical errors (too many to list) throughout the manuscript that should be corrected.

 R = We apologize for the numerous typos and grammar mistakes of the original manuscript. According to this observation, the manuscript has been thoroughly revised and we hope that this reviewer find a significant improvement in the grammar and syntax of the corrected manuscript.________________________________________

6. PLOS authors have the option to publish the peer review history of their article (what does this mean?). If published, this will include your full peer review and any attached files.

Do you want your identity to be public for this peer review? For information about this choice, including consent withdrawal, please see our Privacy Policy.

Reviewer #1: No

Reviewer #2: No

---

## [Decision Letter · Decision Letter 2]

22 Jan 2020

Analysis of the percentages of monocyte subsets and ILC2s, their relationships with metabolic variables and response to hypocaloric restriction in obesity

PONE-D-19-23742R2

Dear Dr. Malacara,

We are pleased to inform you that your manuscript has been judged scientifically suitable for publication and will be formally accepted for publication once it complies with all outstanding technical requirements.

With kind regards,

Melissa M Markofski

Academic Editor

PLOS ONE

Additional Editor Comments (optional):

Reviewers' comments:

Reviewer's Responses to Questions

**Comments to the Author**

1. If the authors have adequately addressed your comments raised in a previous round of review and you feel that this manuscript is now acceptable for publication, you may indicate that here to bypass the “Comments to the Author” section, enter your conflict of interest statement in the “Confidential to Editor” section, and submit your "Accept" recommendation.

Reviewer #1: All comments have been addressed

2. Is the manuscript technically sound, and do the data support the conclusions?

Reviewer #1: Yes

3. Has the statistical analysis been performed appropriately and rigorously? 

Reviewer #1: Yes

4. Have the authors made all data underlying the findings in their manuscript fully available?

Reviewer #1: Yes

5. Is the manuscript presented in an intelligible fashion and written in standard English?

Reviewer #1: Yes

6. Review Comments to the Author

Reviewer #1: The authors have satisfactorily responded to all my questions and made the all necessary changes to the manuscript.

7. PLOS authors have the option to publish the peer review history of their article (what does this mean?). If published, this will include your full peer review and any attached files.

Reviewer #1: No

---

## [Editor Report · Acceptance letter]

30 Jan 2020

PONE-D-19-23742R2 

Analysis of the percentages of monocyte subsets and ILC2s, their relationships with metabolic variables and response to hypocaloric restriction in obesity 

Dear Dr. Malacara:

I am pleased to inform you that your manuscript has been deemed suitable for publication in PLOS ONE. Congratulations! Your manuscript is now with our production department. 

With kind regards,

on behalf of

Dr. Melissa M Markofski 

Academic Editor

PLOS ONE